# Nanoenzyme Reactor-Based Oxidation-Induced Reaction for Quantitative SERS Analysis of Food Antiseptics

**DOI:** 10.3390/bios12110988

**Published:** 2022-11-08

**Authors:** Linmin Chen, Meihuang Zeng, Jingwen Jin, Qiuhong Yao, Tingxiu Ye, Longjie You, Xi Chen, Xiaomei Chen, Zhiyong Guo

**Affiliations:** 1College of Ocean Food and Biological Engineering, Jimei University, Xiamen 361021, China; 2Institute of Analytical Technology and Smart Instruments, College of Environment and Public Health, Xiamen Huaxia University, Xiamen 361024, China; 3College of Pharmacy, Xiamen Medicine College, Xiamen 361005, China; 4National Quality Supervision and Inspection Center for Incense Products (Fujian), Quanzhou 362600, China; 5State Key Laboratory of Marine Environmental Science, Xiamen University, Xiamen 361005, China; 6Xiamen Environmental Monitoring Engineering Technology Research Center, Xiamen 361024, China

**Keywords:** nanoenzyme reactor, oxidation-induced reaction, SERS, food antiseptics

## Abstract

Nanoenzyme reactors based on shell-isolated colloidal plasmonic nanomaterials are well-established and widely applied in catalysis and surface-enhanced Raman scattering (SERS) sensing. In this study, a “double wing with one body” strategy was developed to establish a reduced food antiseptic sensing method using shell-isolated colloidal plasmonic nanomaterials. Gold nano particles (Au NPs) were used to synthesize the colloidal plasmonic nanomaterials, which was achieved by attaching ferrous ions (Fe^2+^), ferric ions (Fe^3+^), nitroso (NO^−^) group, cyanogen (CN^−^) group, and dopamine (DA) via coordinative interactions. The oxidation-induced reaction was utilized to generate •OH following the Fe^2+^-mediated Fenton reaction with the shell-isolated colloidal plasmonic nanomaterials. The •OH generated in the cascade reactor had a high oxidative capacity toward acid preservatives. Importantly, with the introduction of the signal molecule DA, the cascade reactor exhibited also induced a Raman signal change by reaction with the oxidation product (malondialdehyde) which improved the sensitivity of the analysis. In addition, the stable shell-isolated structure was effective in realizing a reproducible and quantitative SERS analysis method, which overcomes previous limitations and could extend the use of nanoenzymes to various complex sensing applications.

## 1. Introduction

Colloidal plasmonic nanomaterials are widely applied in biomedicine [1], catalysis [2], optics [3], electromagnetism, and sensing [4]. The formation of shell-isolated colloidal plasmonic nanomaterials via covalent chemical bonding enables the efficient integration of multifarious bioactive elements. Recently, shell-isolated colloidal plasmonic nanomaterials have been combined with specific modules to simulate reactive oxygen species (ROS) reactions, and thereafter have been effectively applied in oxidation-induced reactions [5]. ROS with high peroxidase or oxidase-mimicking catalytic activities in the shell-isolated structure are sufficient to generate additional ROS such as hydroxyl radicals (•OH) and oxygen radicals (•O_2_^−^) via hydrogen peroxide dissociation [6,7,8]. Further, to significantly decrease the consumption of bio-enzymes in the ROS reaction and enhance the catalytic activity of the oxidation-induced reaction, a series of metal activity-adjusted cascade catalytic systems have been developed. Cheng [9] fabricated a metal organic skeleton-based shell-isolated nanocatalyst and metal-activated cascade system, which exhibited high ROS generation capability owing to a metal oxidase-mimicking catalytic process. Hexahedral Au@AgPt nanoparticles (NPs) with excellent catalytic properties and superior surface-enhanced Raman scattering (SERS) activity were synthesized by Song [10]. These above nanomaterials also exhibited good catalytic reaction activity and better detection behavior for Hg^2+^.

Shell-isolated colloidal plasmonic nanomaterials always exhibit better catalytic properties by the quick transforming of an enzyme matrix to products and accelerating enzymatic kinetics under physiological conditions. Additionally, shell-isolated colloidal plasmonic nanomaterials possess advantageous nanozyme properties [11], structural diversity and stability [12], high catalytic activity [13], good biocompatibility [14], and large-scale-preparation applicability [15]. Consequently, shell-isolated colloidal plasmonic nanomaterials have been widely used as nanoenzymes to mimic the peroxidase/catalase enzymatic activity and replace natural enzymes in photocatalysis [16], diagnosis and treatment [17], sensing [18], and nanoreactors [19]. The shell functionalization of ZIF-67 on Au@Pt bimetallic nanoflowers resulted in the formation of nanoenzymes that could respond in real time to H_2_O_2_. In addition, Au NPs functionalized with AgCl exhibited an enhanced catalytic performance and improved SERS sensitivity [20]. Shell-isolated colloidal plasmonic nanomaterials have also exhibited multifunctional properties when combined with a nanoenzyme system. However, the redox properties have a considerable influence on the target or signal molecules, leading to inaccurate results. Recently, the use of the internal standards (ISs) method has been an effective attempt to improve the reproducibility and quantitative SERS analysis by normalizing volatility of SERS signals [21]. The further integration of ISs and shell-isolated materials is a promising solution to overcome the aforementioned challenges and expand the application of nanoenzyme-based systems in different fields [22].

Food antiseptics are extensively utilized in food engineering to promote food freshness [23]. However, their unregulated use can lead to delayed fetal growth and development [24], gastrohelcosis and damage to the digestive system [25], or fetal malformation. With the development in food science and technology, various antiseptics such as butylated hydroxyanisole [26], butylated hydroxytoluene, tertiary butylhydroquinone, ascorbic acid [27], or polyphenols have been used to hinder the oxidation of certain constituents present in food products. Different types of analytical methods including chromatography [28], electrochemistry [29], and fluorescence spectroscopy [30] have been utilized to analyze antiseptics [31]. Tungkijanansin developed a headspace solid-phase microextraction (HS-SPME) method combined with gas chromatography-flame ionization detector (GC-FID) for the determination of benzoic acid, sorbic acid and propionic acid in fermented food samples with detection limits of 1.1–1.7 mg L^−1^ [26]. The detection method of SA was based on strong SERS of 3,5,5-trimethyloxazole-2,4-dione at 1280 cm^−1^, which was formed by the reaction between malondialdehyde (the oxidation product of sorbic acid) and thiobarbituric acid [31]. Timofeeva developed an air-assisted dispersive liquid–liquid microextraction procedure with the organic phase solidification (AA-DLLME-OPS) method for the detection of benzoic acid and sorbic acid with a detection limit of 0.02 mg L^−1^ by using menthol as an extractant [32]. However, the sensitivity and specificity of current detection methods are insufficient to satisfy food quality evaluation guidelines [33,34]. Therefore, it is necessary to develop novel methods for monitoring food additives with faster, simpler, and highly sensitive properties [35]. Meanwhile, we also observe that antiseptics always exhibit strong reduction which could be utilized in reductive donor oxidation–reduction processes. Furthermore, it is obvious that ROS possesses high oxidization potential to oxide antiseptics. In addition, this reaction is accompanied with a change in the configuration and physical properties of the target [36]. Thus, the combined ROS-generated properties of shell-isolated colloidal plasmonic nanomaterial-based nanoenzymes for food antiseptics sensing are potential candidates to promote food antiseptics monitoring.

In this study, a “double wing with one body” strategy was developed to fabricate reductive food antiseptic sensors using shell-isolated colloidal plasmonic nanomaterials. Au NPs were used as the core of the colloidal plasmonic nanomaterials, and co-assembly with ferrous ions (Fe^2+^), ferric ions (Fe^3+^), the nitroso (NO^−^) group, cyanogen (CN^−^) group, and dopamine (DA) to form the sensors. In the presence of H_2_O_2_, large amounts of •OH were generated by Fenton-like reactions [37] and exhibited excellent peroxidase-like activity, and also catalyzed sorbic acid oxidation to malondialdehyde (MDA). The formed MDA reacted with DA molecule [38] and induced a Raman signal change, which has a clear linear relationship with the concentration of SA. The present work demonstrated that the nanoenzyme significantly enhanced sorbic acid sensing performance and could be used for the examination of sorbic acid concentrations in real food samples.

## 2. Materials and Methods

### 2.1. Materials

All chemicals, including gold(III) chloride trihydrate (HAuCl_4_·3H_2_O, ≥99.9%), sorbic acid (C_6_H_8_O_2_, AR, 99%), sodium citrate (C_6_H_5_O_7_Na_3_, 98%), hydrogen peroxide solution (H_2_O_2_, 30%), benzoic acid (C_7_H_6_O_2_, CP, 99.0%), sodium nitroferricyanide dihydrate (C_5_FeN_6_Na_2_O·2H_2_O, 99.98%), potassium pyrosulfite (K_2_S_2_O_5_, CP, 95%), potassium ferrocyanide trihydrate (K_4_FeC_6_N_6_·3H_2_O, AR, 99.0%), p-hydroxybenzoic acid (C_7_H_8_O_2_, 99%), iron(III) sodium acetate anhydrous (CH_3_COONa, AR), zinc acetate (C_4_H_6_O_4_Zn·2H_2_O, AR, 99%), dopamine hydrochloride (C_8_H_11_NO_2_·HCl, 98%), 3,3’,5,5’-tetramethylbenzidine dihydrochloride (C_16_H_20_N_2_·2HCl, 98%), sodium phosphate dibasic dodecahydrate (Na_2_HPO_4_·12H_2_O, AR, 99%), and chloride hexahydrate (FeCl_3_·6H_2_O), were purchased from Aladdin Chemical Reagent Co., Ltd. (Shanghai, China). All other reagents were at least analytical grade and were used without further purification. Ultrapure water (resistivity up to 18.2 MΩ·cm) was used throughout the experiment.

### 2.2. Preparation of Au@SNF NPs (Sodium Nitroprusside Framework Encapsulated with Gold Nanoparticles)-Based Nanoenzyme Reactor

**Synthesis of Au NPs.** Au NPs were synthesized as follows [39]. First, sodium citrate solution (1 mL of 1% (*w*/*v*)) was added to a micro-boiling HAuCl_4_ solution (100 mL of 0.01% (*w*/*v*)), and was maintained for a few minutes in this state. Subsequently, the temperature of the mixture was decreased slightly and held under a micro-boiling state (40 min). The occurrence of the reduction and annealing reactions was signified by a color change in the solution. Finally, the samples were cooled to room temperature and subjected to centrifugation to obtain Au NPs (size: 45 nm).

**Preparation of Au@SNF NPs.** Au@SNF NPs with a uniform shell layer thickness were synthesized according to our previous method. Briefly, 2 mL of a 0.5 mmol/L sodium nitroferricyanide dihydrate solution was used to pretreat the Au NPs (10 mL) for 15 min to obtain a CN^−^-etched Au NP surface. This surface enabled the efficient integration of multifarious bioactive elements by covalent chemical bonding. Subsequently, the covalent sections containing Fe^2+^, Fe^3+^, CN^−^, and NO were injected into the Au@CN^−^ NPs solution with a speed of 0.1 mL/min and reacted for 1 h. The obtained complex was then centrifuged (8000 rpm/min for 10 min), following which the supernatant solution was discarded and the centrifugation was dispersed in ultrapure water and concentrated (10-fold) to obtain Au@SNF NPs.

**Fabrication of the Au@SNF NPs-based nanoenzyme reactor.** In this nanoenzyme reactor, the Au@SNF NPs contained CN^−^ and Fe sources, which were utilized as the signal internal standard molecule and Fenton reaction generator. In this system, the SERS signal primer DA was covalently bonded around Au@SNF NPs through the self-assembly strategy. The colloidal plasmonic nanomaterial-based nanoenzyme cascade reactor was fabricated in a stepwise manner, where Au NPs and Fe precursors were selected as the vital components of the nanoenzyme cascade reactor. The obtained Au@SNF NPs were dispersed in ultrapure water through sonication. Thereafter, the DA molecules (10^−2^ M) were introduced into the Au@SNF NP solution and maintained for a few minutes to synthesize the nanoenzyme reactor. The nanoenzyme reactor was then stored at 4 °C for future use. Spectroscopic studies were performed to characterize the prepared nanoenzyme cascade reactor.

### 2.3. Sensing Performance of the Au@SNF NP-Based Nanoenzyme Reactor

**Catalytic performance analysis of nanoenzyme reactors.** The catalytic efficiency of the Au@SNF NPs was analyzed by using the 3,3′,5,5′-tetramethylbenzidine (TMB) colorimetric reaction, which is a classical mimetic peroxidase means. The TMB catalytic reaction was carried out in a reaction solution containing H_2_O_2_ (50 μL, 0.03%), TMB (50 μL, 10 mM), and Au@SNF NPs, and the solution was incubated for 5 min at room temperature. The catalytic performance of the Au@SNF NPs was monitored by ultraviolet-visible (UV-vis) spectroscopy at approximately 652 nm, where the oxTMB (oxidized TMB) state was generated.

**Quantitative detection of sorbic acid concentration.** The Au@SNF NPs were used as a nanoenzyme reactor to determine the detection limit of sorbic acid (SA). A Fenton-like reaction was performed by mixing 100 μL of the well-dispersed nanoenzyme reactor and 50 μL of H_2_O_2_ (0.03%) for 2 min to generate •OH. Subsequently, the oxidation-induced reactions were initiated with the addition of 100 μL of SA at various concentrations for 5 min at room temperature. These reactions were expedited by the generated •OH, which accelerated the oxidation of SA to MDA; this in turn reacted with DA and reduced the intensity of the characteristic peak of DA. The change in the signal intensity as a function of the SA concentration was measured using an ATR8300AF AutoFocus micro-Raman spectrometer.

**Selectivity evaluation assay.** Common food preservatives such as benzoic acid (C_7_H_6_O_2_), sodium phosphate dibasic dodecahydrate (Na_2_HPO_4_·12H_2_O), potassium pyrosulfite (K_2_S_2_O_5_), sodium acetate anhydrous (CH_3_COONa), and p-hydroxybenzoic acid (C_7_H_8_O_2_) were selected to evaluate the selectivity of the reaction system. The above food preservatives were tested at a concentration of 10^−4^ mol/L and SA concentration of 10^−5^ mol/L. The selectivity experiments were carried out as described above and analyzed using an ATR8300AF AutoFocus micro-Raman spectrometer.

**Reproducibility evaluation assay.** The reproducibility of the Fenton-like reaction for the determination of SA was assessed through ten experiments. The concentration of SA was maintained constant at 10^−5^ mol/L, and other tests were performed as described here. The experiment was repeated three times.

### 2.4. Actual Sample Testing Using the Au@SNF NP-Based Nanoenzyme Reactor

The practical applicability of the Au@SNF NP-based nanoenzyme reactor was validated for the determination of SA in food samples using a series of common pastries. The food samples were prepared according to the Chinese national standard method (GB 5009.28-2016, Determination of benzoic acid, sorbic acid and sodium saccharin in food). The efficiency of the nanoenzyme reactors was analyzed for SA following the above method without modification. The detailed pre-treatment processes of the food samples are described in the Appendix A.

### 2.5. Characterization of Au@SNF NPs-Based Nanoenzyme Reactor

**Spectrum behaviors study.** The UV-vis (200 nm to 800 nm) and SERS optical behaviors (50 mW laser power, 6000 ms integration time, and 785 nm laser) and the sensing test of the nanoenzyme reactor were studied using a UV-2600i spectrometer (Shimadzu, Kyoto, Japan) and autofocus micro-Raman spectrometer (ATR8300AF, Xiamen, China).

**Morphology characterizations.** Scanning electron microscopy (SEM, Hitachi-4800 (Japan)) and transmission electron microscopy (TEM, Tecnai-G2-F20 (FEI, Portland, OR, USA)) were utilized to characterize the morphology and microstructure of nanoenzyme reactor. The high-angle annular dark-field scanning transmission electron microscopy (HAADF STEM) imaging was used to analyze the composition of the prepared nanoparticles.

**Surface behavior study**. X-ray photoelectron spectroscopy (VG, Hyannis Port, MA, USA) allowed for the characterization of the elemental composition of the material as well as its chemical state and molecular structure. The surface behavior of the Au@SNF NPs was performed with zeta potential measurement (Malvern Nano ZS90, Malvern City, UK).

## 3. Results and Discussion

### 3.1. Characterization of the Sodium Nitroprusside Framework (SNF)-Based Nanoenzyme Reactor

The construction process of the colloidal plasmonic nanomaterials-based nanoenzyme cascade reactor was performed in a stepwise manner (Appendix A), in which Au nanoparticles and Fe precursors and ligands were selected as the vital modules of the nanoenzyme cascade reactor. Firstly, through optimization (Appendix A), Au NPs (45 nm) with stable LSPR properties were finally selected as cores, subjected to etching reactions and thus we obtained Au@CN^−^ NPs. The CN^−^-etched Au NPs could provide connection sites for metal and ligand assembly around the Au@CN^−^ NPs surface. Then, NO^−^, CN^−^ and Fe precursors (Fe^2+^ and Fe^3+^) were connected by metal coordination interactions to form metal a sodium nitroprusside (SNP) complex. Finally, the obtained shell-isolated colloidal plasmonic nanomaterials were reacted with dopamine to fabricate the nanoenzyme cascade reactor. The UV-vis spectroscopy and Raman spectra were analyzed to characterize the prepared nanoenzyme cascade reactor (Appendix A). In the UV-vis spectroscopy (Appendix A), the two localized surface plasmon resonance (LSPR) bands located at 526 nm and 600–800 nm can be assigned to Au NPs and SNF species, respectively [40,41]. Furthermore, with the fabrication of the nanoenzyme cascade reactor, the Raman peaks located in the traditionally Raman-silent region (2185 cm^−1^) were now assignable as CN^−^ Raman peaks (Appendix A). The corresponding LSPR bands and Raman peaks represent the spectral characteristics of the nanoenzyme cascade reactor, confirming the successful preparation of the colloidal plasmonic nanomaterials.

Structural characterization was performed using SEM (Figure 1A,B) and TEM (Figure 1C,D). These results confirmed the spherical structure and determined the average diameter as 40–50 nm. The size was also confirmed by the DLS studies (Appendix A). The TEM and SEM images also confirmed that the nanoenzyme cascade reactor had a well-defined core-shell structure, with an SNP complex shell thickness of 2 nm and a circular crosslinking shell structure. In order to demonstrate the composition of the nanomaterials, we performed an elemental mapping characterization of the Au@SNF NP by HRTEM. The characterization results are shown in Appendix A and presented distinctly Au and Fe elements in Au@SNF NPs. The HRTEM images exhibited Au as the obvious gold core structure and Fe as the obvious SNF skeletal structure, which further confirm the successful preparation of Au@SNF NPs. The HRTEM image (inset of Figure 1D) distinctly presents different lattice space of 0.235 and 0.135 nm, which can be assigned to the (111) plane of Au (JCPDS 04-0784) and (111) plane of SNF, respectively. The atomic constitution and surface-bonding properties of the Au@SNF NPs were determined by X-ray photoelectron spectroscopy (XPS; Figure 2A). The C 1s spectra (Figure 2B) of the Au@SNF NPs show a single peak at 285.2 eV corresponding to the C-N group. The O 1s spectra (Figure 2C) of the Au@SNF NPs also exhibit a single peak at 532.2 eV, which is assigned to the N = O group. Moreover, the N 1s spectra (Figure 2D) of the Au@SNF NPs exhibit two peaks at 399.5 and 398.2 eV, which are assigned to the N = O and C-N groups, respectively. The Au 4f spectra (Figure 2E) show Au 4f7/2 and Au 4f5/2 spin states around 84.0 and 87.9 eV, respectively, which are attributed to the zero-valent states of gold. However, since Fe ions were primarily present in the interior of the Au@SNF NPs rather than on the surface, the peaks of Fe are not significant, and the Fe 2p1/2 and Fe 2p3/2 spin states are detected around 724.5 and 711.2 eV, respectively, in the Fe XPS profiles (Figure 2F). These results confirm the successful preparation of the Au@SNF NPs through metal coordination. The surface behavior of the Au@SNF NPs was analyzed by zeta potential measurements (Appendix A). The zeta potential of the Au@SNF NPs was the main factor enabling their interaction with the target and facilitating the self-assembly of the NPs. The zeta potential results confirm that the Au@SNF NPs are positively charged (ζ = 18.5 mV) owing to their Fe-covalent surface. The coordination through NO-Fe-NO, CN-Fe-CN, and NO-Fe-CN resulted in the formation of an SNF structure around the AuNPs, which was caused by the excellent complexion interaction of Fe atoms.

### 3.2. Peroxidase-like Catalytic Activity of Au@SNF NPs-Based Nanoenzyme Reactor

The core-shell colloidal plasmonic nanomaterials (Au@SNF NPs) consisted of a sodium nitride framework with shell-like structures, CN^−^ ISs, specific NO groups, and Fe-covalent surfaces that exhibited excellent substrate effects and suitability for SERS analysis. The nanoenzyme reactor was constructed by combining the signal molecule DA with Au@SNF NPs. The absorption spectra of all nanomaterials are shown in Appendix A; this reactor exhibited peroxidase-like catalytic activity because of the Fe source and the DA molecule. To evaluate the peroxidase-like catalytic behaviors of the nanoenzyme reactor, we investigated these different responses and the results are shown in Figure 3. Firstly, the reactive oxygen species (ROS) generation by the Fenton-like reaction of the nanoenzyme reactor with an analyte-activated catfish was investigated using the TMB-H_2_O_2_ chromogenic system. The UV-vis spectroscopy and colorimetric responses of different reaction systems were analyzed and the results are shown in Figure 3A. The UV-vis spectroscopy for the nanoenzyme-TMB and nanoenzyme-H_2_O_2_ system showed no obvious color change. The H_2_O_2_-TMB system exhibited improved catalytic activity compared to the other two, marked by its weak UV-vis spectroscopy (at 652 nm) and negligible color change, thus indicating restricted oxTMB production. In contrast, the nanoenzyme-H_2_O_2_-TMB system showed significant color response and UV-vis spectroscopy variation; its absorbance intensity (at 652 nm) was 12 times higher (Appendix A) than that of the H_2_O_2_-TMB reaction system. In addition, no ROS were generated and no obvious color changes were noticed for the other reaction systems. These results were attributed to the ability of the nanoenzymes (with a high degree of localization and plasmonic effects) to react with H_2_O_2_ to produce a high Fenton-like catalyticity, which generated •OH and •O_2_^−^, and oxidized TMB to oxTMB at a higher rate. On the other hand, no ROS were generated and no obvious color changes were noticed for the other reaction systems.

Efficiency of the Fenton-like reaction with different parts of the nanoenzyme reactor was investigated. The DA part of the reactor always exhibited an enhanced Fenton-like reaction effect compared with the other parts. We also investigated the UV-visible absorption properties and colorimetric responses of the nanoenzyme reactor in the presence of DA. From the results, it was apparent that the H_2_O_2_-Fenton-like reaction system exhibited strong UV-vis spectroscopy at 652 nm (curve d) and the intensity of the peak was 1.6-times higher than that for the single Au@SNF NPs (Appendix A). In this study, Au@SNF@DA was used as the catalyst to elucidate the potential Fenton-like reaction mechanisms (Figure 3D). In brief, in the nanoenzyme reactor environment, the Fe^3+^ ions were reduced to Fe^2+^ ions due to reducibility of the DA molecule (Figure 3D (1)). Catalyzed by Au@SNF@DA, H_2_O_2_ was reduced by Fe^2+^ and oxidized by Fe^3+^ to produce •OH radicals and •O_2_^−^ molecules, respectively (Figure 3D (2) and (3)). Then, Fe^2+^ ions reacted with •O_2_^−^ molecules to generate abundant •O_2_^−^. Finally, the TMB reacted with the •OH and •O_2_^−^ radicals to generate the excited-state oxidation product, i.e., oxTMB, which is characterized by a strong UV-vis spectroscopy and color change. The DA molecule triggered a strong Fenton-like reaction and enhanced the reduction of Fe^3+^ to Fe^2+^, and the H_2_O_2_ available in the system initiated the successive generation of •OH and •O_2_^−^ radicals.

In addition, the apparent steady-state kinetics of the nanoenzyme reactor were evaluated using different concentrations of TMB and H_2_O_2_ (Appendix A). The nanoenzyme reactor catalyzed the typical Michaelis–Menten reaction between hydrogen peroxide and the TMB system (Figure 3C and Appendix A, respectively). The data were fitted by the Michaelis–Menten equation to obtain the maximum reaction velocity (V_max_) and the Menten constant (K_m_, which denotes the affinity of the enzyme for the substrate) [42,43]. The V_max_ of the nanoenzyme reactor for the TMB substrate was determined to be 1.6 × 10^−8^ M s^−1^ and the K_m_ value of the nanoenzyme reactor was found to be 0.265 mM (Figure 3C). These results indicate that the nanoenzyme reactor showed excellent affinity and catalytic activity for the substrate TMB.

### 3.3. Construction of a Sensing Method System by Au@SNF NPs-Based Nanoenzymes

In this study, the above-mentioned nanoenzyme reactor exhibited excellent affinity and catalytic activity for the substrate TMB and led to successive generation of •OH and •O_2_^−^ radicals. In previous reports, ROS offer a route to oxidize potential and always result in transformation of reductive antiseptics structure which may be accompanied by configuration or characteristics. Moreover, considering the enzymatic activities of Au@SNF NPs-based nanoenzymes, it was believed that oxygen-based radicals could oxidize the reducibility-related molecules. For this purpose, a classic cascade MDA reaction was introduced to verify the application of the Au@SNF NP-based nanoenzymes. The related mechanism for the Fenton-like effect with an analyte-activated catfish and MDA reaction is shown in Figure 4A. Overall, the self-redox properties of the Au@SNF NP-based nanoenzymes induced by H_2_O_2_ were particularly crucial for continuous generation of •OH and •O_2_^−^ radicals, and the oxidation of SA molecules occurred sequentially. The highly localized, hot-electron and thermal-plasmon effects of the LSPR effect resulted in a dramatic enhancement of the Raman signal with molecules located around nanomaterials. When the signal molecule DA was introduced in the nanoenzyme, the ‘fingerprint’ SERS signal was enhanced (at 578, 813, 1258, 1313, 1419 and 1463 cm^−1^), which can be used for the validation of the MDA reaction. The generated MDA eventually combined with the DA molecules in Au@SNF NPs and resulted in prominent changes to the SERS signals. To study this, an oxidation–reduction cascade reaction was designed to correlate the SA concentration indirectly.

To validate the MDA-forming reaction, MDA test kits were introduced in the nanoenzyme reactor. MDA is a secondary mutagenic product of polyunsaturated fatty acid peroxidation and is commonly used as a food oxidation product. The transformation process of SA to MDA is always accompanied by a color change and an absorption peak at 532 nm. After the introduction of MDA test kits, the SA and Au@SNF NP-based nanoenzyme system exhibited obvious color change and a strong absorption peak at 532 nm; on the other hand, Au@SNF NPs did not show any UV activity in the absence of SA (Figure 4B). The production of •OH and •O_2_^−^ radicals triggered the transformation between SA and MDA. Interestingly, the Au@SNF NPs-H_2_O_2_ system showed a significant color change and an intense absorption peak compared to the Au@SNF NP-based nanoenzymes system. The Au@SNF NP-based nanoenzymes system generated oxTMB, which in turn resulted in the formation of more •OH and •O_2_^−^ radicals. In the cascade reaction, MDA was consumed by the DA molecules. Figure 4C shows the SERS behaviors of the Au@SNF NPs and the Au@SNF NP-based nanoenzymes with cascade reaction. These results showed that the SERS signal decreased in intensity upon adding the SA molecules. These phenomena indicated that the oxidation–reduction cascade reaction occurred and can be used for the rapid detection of SA molecules. In contrast to the reagent kit reaction, which requires high temperatures (100 °C) and long reaction times (1 h), the assay system suggested herein requires only 7 min at room temperature to detect MDA, which is important for practical rapid detection applications. Overall, the combination of DA autoxidation and Fenton-like reactions resulted in the generation of large numbers of ROS, which enabled the nanoenzyme reactor to improve the efficiency of MDA generation, thus significantly lowering the detection limit and improving the detection sensitivity.

### 3.4. Oxidation–Reduction Cascade Reaction for SA Detection

SA is an important food antiseptic and is used extensively in food preservation processing. Thus, the antiseptic-related reduction behaviors are worthy of attention. The above-mentioned oxidation–reduction cascade reaction can be used as a novel method for SA determination. The SERS signal of the detection system originated from the nanoenzyme reactor subjected to a convective mediation at a constant DA characteristic peak. To obtain accurate SERS results, the stability of Au@SNF NPs was examined and the results are shown in Appendix A, where Au@SNF NPs have good stability. Thus, Au@SNF NPs with the characteristic peak (2185 cm^−1^) were recorded and used as an internal standard to correct Raman fluctuations. The changes in the SERS signal from the nanoenzyme reactor with increasing SA concentration were recorded, and the Raman signal intensity was found to decrease gradually, indicating that the nanoenzyme reactor catalyzed a significant SA-content-dependent SERS response in the reaction system. Under the optimized conditions (Appendix A), the SERS intensity ratio of 585 cm^−1^/2185 cm^−1^ had a good linear correlation with the logarithm of SA concentration in the range from 100 nM to 1 mM (Figure 5A), and the limit of detection (LOD) was as low as 12 nM (Figure 5B). The detection limit of the nanoenzyme reactor fabricated in this study was significantly lower than the maximum allowable use of SA in food (0.075–2 g/kg). Therefore, the sensing method based on the nanoenzyme reactor can be used to investigate the SA concentration in real food samples.

To evaluate the selectivity of the nanoreactor-promoted Fenton-like reaction system for SA detection (via SERS method), different types of food preservatives, such as sodium dihydrogen phosphate, sodium acetate, benzoic acid, p-hydroxybenzoic acid, and potassium metabisulfite, were added as interfering substances. The interfering substance (10^−4^ M) signal was similar to that of the blank solution, which confirmed the high specificity of this method (Figure 5C). This was because the Fenton-like reaction dominated by the nanoenzyme reactor intensified the oxidation of SA to MDA, while the combination of MDA and DA destroyed the combined structure of the nanoenzyme reactor and reduced the reactor signal, thereby exhibiting better selectivity for SA. Moreover, the selectivity for SA detection depended on the role played by different parts of the sensing method.

To ensure that the method can be used to accurately quantify SA, the reproducibility and homogeneity of the method was measured. For this purpose, we randomly measured 10 sets of SA solutions (10^−5^ M) under the same reaction conditions and plotted the SERS intensity ratio of 585 cm^−1^/2185 cm^−1^ (Figure 5D). After data processing, the relative standard deviation (RSD%) of the corrected 585 cm^−1^ peak was found to be 1.75%, indicating the high reproducibility and homogeneity of the method. The Au@SNF-based nanoenzyme reactor can solve the problem of low reproducibility of the SERS substrate and meet the requirements of quantitative analysis. The method could be used for real-time quantitative analysis in different fields.

### 3.5. Sample Analysis

Previously, we demonstrated the excellent performance of the SA detection platform constructed with a sensing method, and the aim of this study was to identify the capability and feasibility of the method in terms of practical applications. To further explore the practical applications of this nanoenzyme-based SA detection system, SA in food (cakes) was tested. At first, the samples were pretreated following the national standard method of China (GB 5009.28-2016 determination of benzoic acid, SA, and sodium saccharin in food) and were analyzed using the nanoenzyme reactor. The intensity ratio of the peaks at 585 and 1285 cm^−1^ was converted to SA content, based on the SA calibration curve. As shown in Table 1, the SA content of the actual samples was determined to be less than the maximum usage limit of 1.0 g/kg for confectionery, which was in accordance with the national limit (GB 5009.28-2016 determination of benzoic acid, SA, and sodium saccharin in food). The RSD of the method based on the nanoenzyme reactor was 0.37–2.37%, and the recoveries obtained ranged from 90.24% to 109.40%; compared to other sorbic acid detection methods (Appendix A), these data indicate the application potential of the method in the determination of complicated food samples. In summary, these results confirmed that the detection system based on the nanoenzyme can be used to determine the concentration of SA in real food samples.

## 4. Conclusions

In summary, we proposed an oxidation-induced reaction for the detection of food preservative SA in complex food products with SERS. The shell-isolated colloidal plasmonic nanomaterials Au@SNF NPs-DA system was selected as a nanoenzyme to generate •OH and •O^2−^ with Fenton-like reactions. The •OH generated in the cascade reactor had a high oxidative capacity toward acid preservatives, especially for SA. The formed MDA reacted with the DA molecule and induced a Raman signal change, which has a clear linear relationship with the concentration of SA. The as-mentioned method exhibited high sensitive and satisfactory recoveries for SA determination in real food sample assays. This novel sensing method has great potential for the quantitative analysis of preservatives in complex food products. Furthermore, the use of IS molecules in the shell-isolated structure was effective in realizing a reproducible and quantitative SERS analysis method, which overcomes previous limitations and could extend the use of nanoenzymes to various complex sensing applications.

## Figures and Tables

**Figure 1 biosensors-12-00988-f001:**
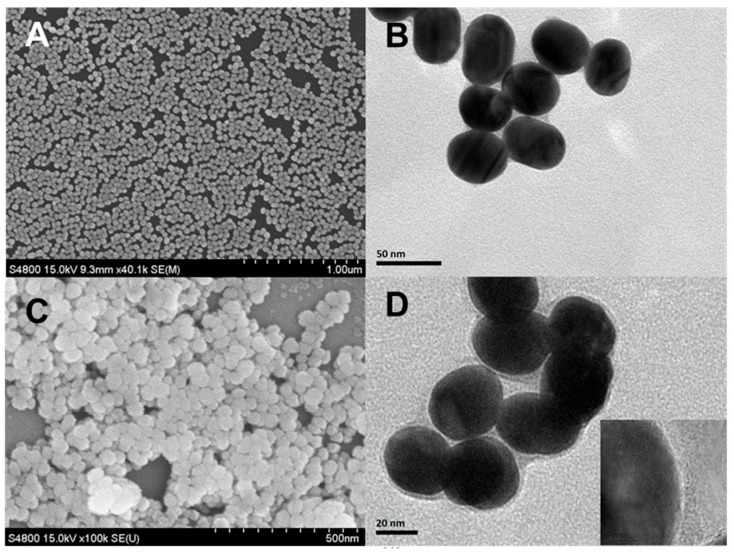
SEM images of the Au NPs (**A**) and Au@SNF NPs (**B**), TEM images of the Au NPs (**C**) and Au@SNF NPs (**D**), HRTEM images of the Au@SNF NPs ((**D**), insert).

**Figure 2 biosensors-12-00988-f002:**
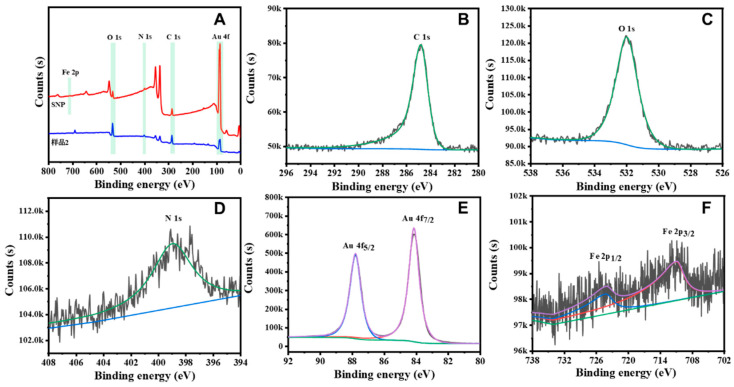
XPS pattern of Au@SNF NPs (**A**), High resolution XPS spectra of (**B**) C 1s, (**C**) O 1s, (**D**) N 1s, (**E**) Au 4f, (**F**) Fe 2p.

**Figure 3 biosensors-12-00988-f003:**
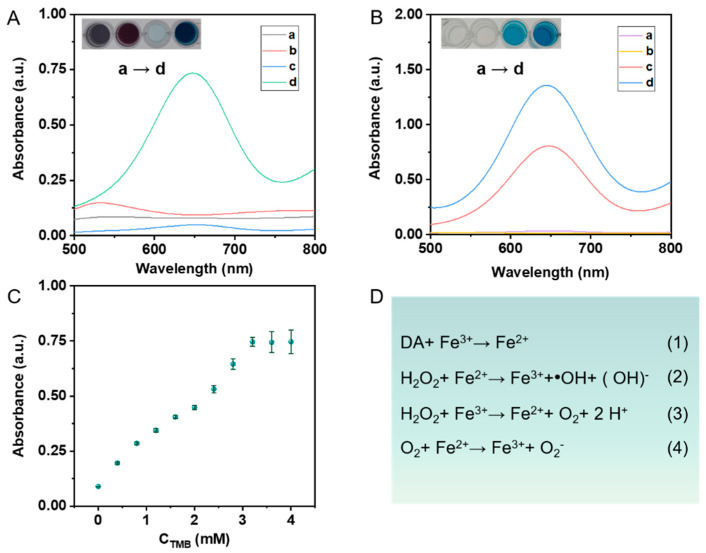
(**A**) UV-vis absorption spectral and color changes of the reaction systems: (a) without H_2_O_2_, (b) without TMB, (c) without nanoenzyme reactors, (d) nanoenzyme reactors-H_2_O_2_-TMB. (**B**) UV-vis absorption spectral response and color changes of the Fenton-like reaction: (a) without H_2_O_2_, (b) without TMB, (c) Fe^3+^-H_2_O_2_-TMB, (d) Fe^3+^-DA-H_2_O_2_-TMB. (**C**) The steady-state kinetic assay of Au@SNF NPs-catalyzed oxidation of TMB by H_2_O_2_. (**D**) The mechanism of the peroxidase-like catalytic ability with Au@SNF NPs.

**Figure 4 biosensors-12-00988-f004:**
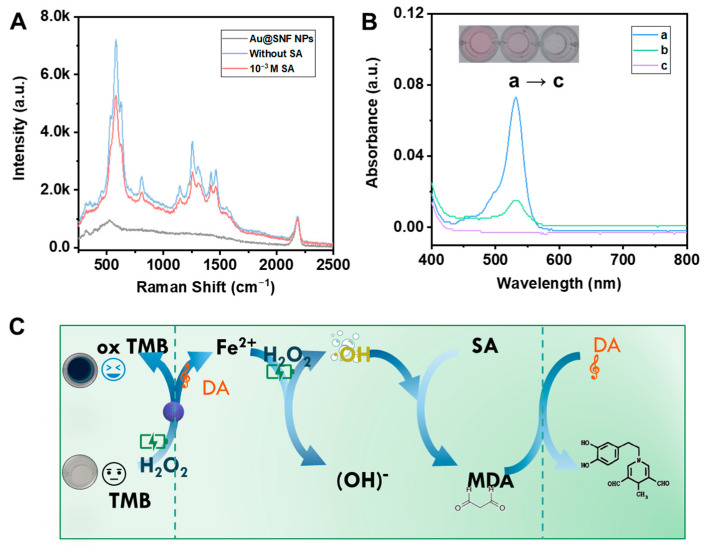
(**A**) The SERS behaviors of the nanoenzyme reactor for detecting SA. (**B**) The UV-vis absorption spectra: (a) Au@SNF NPs/H_2_O_2_/malondialdehyde test kit/SA, (b) Au@SNF NPs/DA/H_2_O_2_/malondialdehyde test kit/SA, (c) Au@SNF NPs/H_2_O_2_/malondialdehyde test kit. (**C**) The mechanism of the nanoenzyme reactor for detecting SA.

**Figure 5 biosensors-12-00988-f005:**
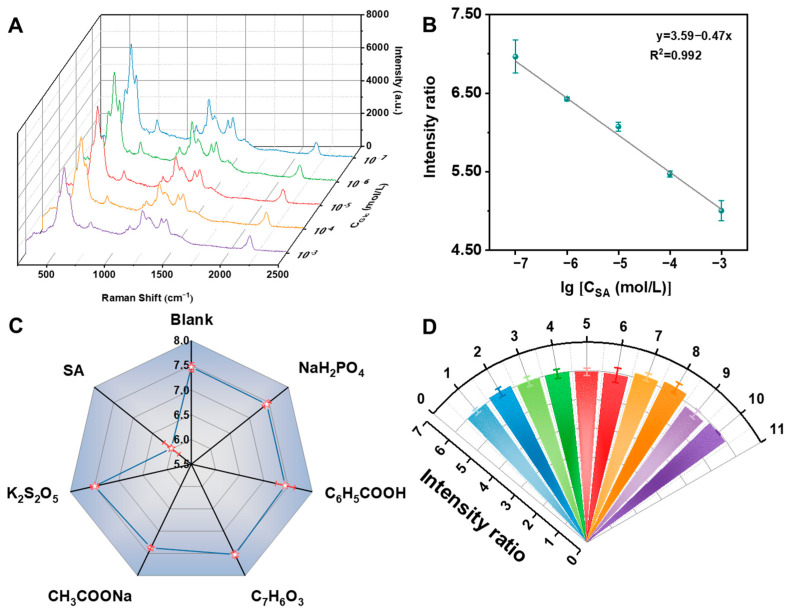
(**A**) The SERS spectra of this system at various concentrations of sorbic acid (10^−3^–10^−7^ mol/L). (**B**) Calibration plot based on the Raman intensity of 585 cm^−1^/2185 cm^−1^. (**C**) The SERS response of 585 cm^−1^/2185 cm^−1^ to 10 µM of sorbic acid and 0.1 mM others. (**D**) The SERS intensity ratio of 585 cm^−1^/2185 cm^−1^ obtained from 10 sets and measured three times in parallel with 10 μM glucose solution.

**Table 1 biosensors-12-00988-t001:** The results of SA count with the nanoreactor-based sensing method.

Samples	This Method (g/Kg)	RSD (%)	GB 2760-2014	Added (g/Kg)	Recovery (%)	RSD (%)
cake-1	0.21	0.37	≤1 g/Kg	1	93.51	1.39
cake-2	0.13	1.04	≤1 g/Kg	1	105.82	1.13
cake-3	0.84	2.37	≤1 g/Kg	1	109.40	2.16
cake-4	0.08	1.58	≤1 g/Kg	1	90.24	1.07

## Data Availability

Not applicable.

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
