# Peer review of "Nanoenzyme Reactor-Based Oxidation-Induced Reaction for Quantitative SERS Analysis of Food Antiseptics"

_biosensors, 2022, doi:10.3390/bios12110988_

Round 1

Reviewer 1 Report

The manuscript has been well written in simple english. I request the authors to check for spelling and grammer mistakes. I am satisfied with the manuscript.

Author Response

Reviewer 1

  1. The manuscript has been well written in simple English. I request the authors to check for spelling and grammar mistakes. I am satisfied with the manuscript.

Answer: We appreciate it very much for the reviewer point out our errors in English. We have checked the manuscript carefully, and the phrase errors have been corrected in the revised version. These revisions are highlighted in the revised manuscript.

Reviewer 2 Report

In this study, employing shell-isolated colloidal plasmonic nanoparticles, reductive food antiseptic sensors were fabricated using a "double wing with one body" method. Gold nanoparticles (Au NPs) were utilized to create colloidal plasmonic nanomaterials, and co-assembly was achieved via coordinative interactions between ferrous ions (Fe2+), ferric ions (Fe3+), nitroso (NO-) group, cyanogen (CN-) group, and dopamine (DA). It should undergo revision before this work can be published in a scientific journal.

1.     What is the reason for using the AuNPs as the plasmonic material? What are the other plasmonic materials? What are those merits and limitations?

2.     What are the reason for using the 45 nm of AuNPs? Authors should add the histogram of TEM images. What are the effect of diameter on the performance? 

3.     Authors should make some schematic of synthesis of nanozyme.

4.     Authors should also do the TEM-EDX of synthesized nanomaterials to showcase the material composition. 

5.     Authors should also show the absorbance spectrum of all synthesized nanomaterials. 

6.     What are the reasons for using the several noise in the Figure 2?

7.     Authors should add some more performance analysis results.

8.     Authors should also compare the performance with existing models in a tabular form. 

Author Response

Reviewer 2

  1. What is the reason for using the AuNPs as the plasmonic material? What are the other plasmonic materials? What are those merits and limitations?

Answer: Thanks for your kind suggestion. Au NPs, as the most classical SERS substrates, have excellent tenability, stability, biocompatibility and catalytic properties, which are widely used in cellular analysis, drug delivery, cell/tissue imaging and quantitative analysis. In hence, Au NPs always selected as the core of colloidal plasma nanoparticles and used for the quantitative analysis. In addition, Ag NPs are also plasmonic material and commonly used in SERS analysis. Ag NPs have strong SPR effects and exhibited with SERS analysis sensitivity enhanced properties. However, Ag NPs exhibited with poor stability and susceptible to environmental changes which makes it unsuitable as SERS substrate. And that, in our work, we utilized etching reaction to modify colloidal plasma nanomaterials which always caused Ag NPs structure collapse serious and disassembly. In hence, we performed Au NPs as candidate, formed the core-shell structure and executed the SERS analysis.

  1. What are the reason for using the 45 nm of AuNPs? Authors should add the histogram of TEM images. What are the effect of diameter on the performance?

Answer: Thanks for your kind suggestion. Au NPs, as the core of shell-isolated nanoparticles, its particle size exhibited with larger influence on the SERS effect. The suitable nanoparticle size can ensure the uniform distribution of CN- on the surface of Au NPs, maintain the structural integrity and avoid collapse of Au NPs. In hence, we have compared different size (15 nm, 45 nm and 60 nm) in detail and result showed in Figure S2. As shown in Figure S2, the strongest signal of Au@SNF NPs was emerged with 45 nm Au NPs. And that, the 45 nm Au NPs always exhibited with enhanced SERS signal than the other which are advantage in related analysis.

Figure S2. Optimization of the size of Au NPs.

In addition, we have provided the histograms of TEM images of Au NPs and Au@SNF NPs (Figure S4).

Figure S4. The histograms of the Au NPs (A) and Au@SNF NPs (B).

  1. Authors should make some schematic of synthesis of nanoenzyme.

Answer: Thanks for your kind suggestion. The schematic of nanoenzyme synthesis is provided in following. Firstly, through optimization (Figure S2), Au NPs (45 nm) with stable LSPR properties were finally selected as cores, subjected to etching reactions and obtained the Au@CN- NPs. The CN- etched Au NPs could provide connection sites for metal and ligand as-sembly around Au@CN- NPs surface. Then, NO-, CN- and Fe precursors (Fe2+ and Fe3+) was connected by metal coordination interaction and formed the metal sodium nitro-prusside (SNP) complex. Finally, the obtained shell-isolated colloidal plasmonic na-nomaterials were reacted with dopamine to fabricate the nanoenzyme cascade reactor.

Figure S1. Schematic diagram of the nanoenzyme preparation process.

  1. Authors should also do the TEM-EDX of synthesized nanomaterials to showcase the material composition.

Answer: Thanks for your kind suggestion. In order to demonstrate the composition of the nanomaterials, we performed an elemental mapping characterization of the Au@SNF NP by HRTEM (Figure S5). The results exhibited that the existed Au and Fe elements in the Au@SNF NPs which showed core-shell structure. In the HRTEM images, the Au elements as the obvious core structural and Fe as the SNF skeletal structure, which further confirming the successful preparation of Au@SNF NPs.

Figure S5. The HRTEM images and STEM-EDS elemental analysis of Au@SNF NPs.

  1. Authors should also show the absorbance spectrum of all synthesized nanomaterials.

Answer: Following the advice of the expert reviewers, we have added the absorption spectra of all synthesized nanomaterials in Figure S8. The nanoenzyme cascade reactor showed similar UV-vis absorbance spectrum around 526 nm and 600-700 bands. And that, the UV-vis absorbance spectrum also demonstrate the different spectrum behaviors towards Au NPs.

Figure S8. The UV-vis absorbance spectrum and colour of different nanomaterials: (a) Au NPs, (b) Au@SNF NPs, (c) Au@SNF NPs-DA.

  1. What are the reasons for using the several noise in the Figure 2?

Answer: Thank you for your question. The Fe ions, N=O and C-N groups were mainly present in SNF skeleton structure. And that, the Au was the mainly elemnt of Au NPs. In hence, in order to verify the SNF skeleton structure existed, we have utilized as the Au, C, O, Fe and N as the represented element to study the nanomaterials.

  1. Authors should add some more performance analysis results.

Answer: Thanks for your kind suggestion. In our work, the performance analysis of stability of Au@SNF NPs, the detection limit of the nanoenzyme reactor, the linear correlation with the logarithm of SA concentration and the high specificity of this method were provided. In order to verify the performance of the method, we examined the stability of the material by random testing of the Au@SNF NPs and plotted the distribution with the peak intensity of the material at 2190 cm-1.

Figure S7. The SERS response of 2190 cm-1 to Au@SNF NPs with ten times.

  1. Authors should also compare the performance with existing models in a tabular form.

Answer: Thanks for your kind suggestion. The SERS assay designed in this work was compared with other methods and the results shown in the table S1. The presesnted method has a wide linear range and low detection limits while compared with the others, which indicating that the method can be used for the sensitive detection of sorbic acid.

Table S1. Comparison of the analytical performance of various determination methods for sorbic acid

Methods

Linearity range

(mg L-1)

Limit of detection

(mg L-1)

recovery percentages

Reference

HS-SPME

5-150

1.1-1.7

92-106%

[1]

AA-DLLME-OPS

0.05-100

0.02

96-101%

[2]

SPE

1-45

0.502

81-96%

[3]

LVSS-PS-CDEKC

0.5-0.01

0.001

97-102%

[4]

SERS

0.0066-1.59

0.0032

95.8-104%

[5]

SERS

0.000112-1.12

1.344×10-5

90.24-109.40%

This work

[1]   Tungkijanansin N, Alahmad W, Nhujak T, et al. Simultaneous determination of benzoic acid, sorbic acid, and propionic acid in fermented food by headspace solid-phase microextraction followed by GC-FID. Food Chemistry. 2020, 329: 127161. 10.1016/j.foodchem.2020.127161.

[2]   Luo Y, Jing Q, Li C, et al. Simple and sensitive SERS quantitative analysis of sorbic acid in highly active gold nanosol substrate. Sensors and Actuators: B. Chemical. 2018, 255: 3187-3193. 10.1016/j.snb.2017.09.144.

[3]   Timofeeva I, Kanashina D, Stepanova K, et al. A simple and highly-available microextraction of benzoic and sorbic acids in beverages and soy sauce samples for high performance liquid chromatography with ultraviolet detection. Journal of Chromatography A, 2019, 1588: 1-7. 10.1016/j.chroma.2018.12.030.

[4]   Kefi B B, Baccouri S, Torkhani R, et al. Application of Response Surface Methodology to Optimize Solid-Phase Extraction of Benzoic Acid and Sorbic Acid from Food Drinks. Foods, 2022, 11(9): 1257. 10.3390/foods11091257.

[5]   Pieckowski M, Kowalski P, BÄ…czek T. Combination of large volume sample stacking with polarity switching and cyclodextrin electrokinetic chromatography (LVSS-PS-CDEKC) for the determination of selected preservatives in pharmaceuticals. Talanta, 2020, 211: 120673. 10.1016/j.talanta.2019.120673.

Reviewer 3 Report

In this experimental paper the authors present their work on the fabrication and practical characterization of a nanoenzyme-based cascade reaction sensing system. To this purpose core-shell nanoparticles (shell-isolated colloidal plasmonic nanomaterials) were fabricated based on gold NPs and functionalized by ferrous ions, ferric ions  (Fe3+), nitroso group, cyanogen group and dopamine. The nanoenzyme oxidation reaction in  the shell-isolated colloidal plasmonic cascade procedure in the presence of hydrogen peroxide  generated hydroxyl radicals following the Fe2+-mediated Fenton reaction. The ·OH group in its turn strongly oxidized different acidic food antiseptics. Dopamine was further introduced as signal molecule, thus ensuring the detection of different acid antiseptics and enabling reproducible and quantitative SERS analysis that goes beyond the state of the art.

The paper is of high quality, relevant, scientifically sound, contains significant contribution and is fully within the scope of the journal. The subject is modern, popular and potentially of a great value in real life applications (detection of excess antiseptics in food). The language is rather good, albeit the style could be better and there are unclear sentences and some mistakes. The text organization and clarity could be improved and some parts should be rewritten. Some nomenclature is rather arbitrarily used and elucidations from a physicist’s point of view (mostly regarding plasmonics) are lacking, while the chemical side is very well presented. The text length is fully suitable to the contents. The supplementary data, given in a separate file, are appropriate and useful. All the experiments are well planned and properly described, thus facilitating their potential reproduction by other teams. References are mostly well chosen. No ethical issues were found when reviewing the text. It is the reviewer’s opinion that the paper will be acceptable for publication after the existing issues have been dealt with in a major revision. The paper has potentials to be well cited upon its publication.

REVIEWER’S POINT-BY-POINT SUGGESTIONS

1. The novelty of the paper over the state of the art should be better stressed in Introduction. This is a very popular and modern topic and a vast number of relevant articles has been published in recent years. This suggestion means that the particular unsolved problem of interest should be fully stated, followed by the proposed approach to overcome them. The paper has its important novelty and the lack of stressing it sufficiently only can work to the authors’ disadvantage.

2. The authors use the term “sensor” to describe their process (nanomaterials and nanoenzyme reactions with the analyte) instead of assigning it to a particular device. Thus “sensing method” would be more appropriate than “sensor”. In a similar vein I also believe that “shell-isolated colloidal plasmonic nanomaterials” would be more up to the point than “shell-isolated colloidal plasmonic cascade reactor”. Please define “sensor” and “reactor”/“nanoreactor” if you use them with specific meanings differing from the generally accepted one. Particularly, I googled “plasmonic cascade reactor” and found zero (!) hits. An identical result has been obtained when searching for “plasmonic cascade nanoreactor”. Usually “sensor” and “reactor” denote devices, not procedures or processes, while “nanoreactor” is a nanostructure serving as a minuscule chemical reactor. It appears that here nanoenzymes and generally core-shell nanoparticles are considered to be sensors and devices on their own.

3. A part of Abstract should be rewritten. The sentence “Importantly, with the introduction of DA as a signal molecule in  the shell-isolated colloidal plasmonic nanomaterials, which reacted with the oxidation products of the acid preservatives while maintaining the stability of the shell-isolated structure” is unfinished and unclear.

4. An additional figure presenting a nanoreactor and a block-scheme of its general design procedure would be beneficial for the overall text clarity.

5. The manuscript is lacking a more explicit description of the physics of localized plasmon influence to the performance of the proposed nanoreactor (primarily extreme light localization, but also hot electron effects and thermoplasmonic effects should be mentioned) within the context of the designed catalytic and SERS activities.

6. The authors repeatedly use the expression “double wing with one body” without explaining it. I guess they meant that their structures can serve two purposes at the same time, namely catalytic/nanoenzymatic and localized plasmonic (SERS-enabling). However, a definition should be explicitly written.

7. Almost the whole last paragraph of Introduction actually gives a summary of the paper, which would be much better suited to Conclusion. It is more customary to just announce the novelty of the paper and (shortly) the methodology used, followed by a single-paragraph description of the paper organization and structure.

8. In line 62, the term “internal standards (ISs)” is used without defining it.

9. Hydroxyl radicals ae written in two different manners, as ·OH and •OH. Is that on purpose? Also, gold nanoparticles are written in two different manners, as “Au NP” and “AuNP”.

10. Some sentences are plain unclear. For instance, what does it mean “generate signals through the self-assembly strategy”? Can “using 3,3',5,5' -tetramethylbenzidine (TMB) mimetic activity of peroxidases classical means” be rewritten in a clearer manner? Other examples include “the SNP complex was assembled around the CN- etched AuNPs and interacted with each other around the monodisperse CN- connected sites”, “spectroscopy behaviors were performed”, “strong Fe atom combined properties”. The clarity and impact of the sentence in lines 209-211 would benefit from either additional explicatory text or by a reference. Please rewrite the unclear sentence “The SERS response of 585 cm-1/2185 cm-1 to 10 µM of glucose with ten times.” in Fig. 5 caption. Please clarify “The method could be used for quantitative analysis and applied for real-time quantitative analysis.” Also clarify “the determination of actual food samples.”

11. Misleading nomenclature for optical spectral properties is used. “UV-visible absorption” is mentioned at 652 nm (e.g. line 284), “UV-vis spectrum” (line 209), “UV-visible spectral behavior” (lines 287, 296), “UV-visible absorption peak” (lines 298, 307, 324, 326, 372, etc.) although the plasmon resonances appear solely in the visible spectrum. A distinction must be made between the material properties (absorption and generally spectral properties which are in the visible) and the equipment and methods used in characterization procedures (which is always UV-vis spectroscopy/spectrophotometry).

12. Some dispersion spectra are given in dependence of wavelength while others are given in dependence on wave vectors. A unified notation would facilitate intercomparison throughout the text for the readers.

13. The abbreviation SNF (Sodium nitroprusside framework) is only defined at 4th page, after being previously used several times. I do not see that the term TMBox is defined anywhere in the text. All abbreviations must be defined the first time they are used.

14. As mentioned in the preamble to these suggestions, the text is mostly well written, meaning that its English is for the most part rather fluent. However, there is also a number of mistakes, erroneous and unclear sentences. Besides the mentioned “sentence” from the Abstract (see suggestion #3) and unclear declarations (suggestion #10), examples of sentences that should be corrected include “sufficient to generated additional ROS” (generate?), “Cheng et al. [9]. fabricated” (point before “fabricated” to be deleted?), “which exhibited high ROS generation capability owing to a metal oxidase-mimicking catalytic process Hexahedral Au@AgPt nanoparticles (NPs) with excellent catalytic properties” (point missing before “Hexahedral”?), “, which also exhibited good  catalytic” (“. These NPs also exhibited good  catalytic”?), “These above materials” is written at the beginning of a paragraph, “catalysis properties” (catalytic?), “their unregulated use can lead to delayed growth and development” (growth and development of what? fetus?), “novel methods  that are faster, simpler, and highly sensitive  for monitoring  food additives” (is this supposed to mean “novel methods  for monitoring  food additives that are fast, simple, and highly sensitive”?), “susceptible by the environment” (?), “reductive food antiseptic sensors” (reductive sensing of food antiseptics?), “precursors and organic-ligand was connected” (were?), “Fe is primarily present in the interior of the Au@SNF NPs than on the surface”, “the resulted are depicted in Figure” (results?), “in the absence and presence of” (=with and without?), “In Hence”, “as interfering substances” and “as the interfering substance.” These are not the only language errors and the manuscript would benefit from critical reading by a native speaker or language expert.

15. The style of the text could be improved by simplifying and rearranging it. A contribution of a language expert would be beneficial here too.

16. Fig. 3A, B, C uses sans serif type of font, while D uses serif font. Kindly use sans serif font (Arial? Helvetica?) in 3 D too.

17. Kindly provide reference(s) for the sentence in lines 333-335.

18. Table 1 is unreferenced in the text.

19. Conclusion appears unfinished. It reiterates the text written at the end of Introduction and skips the real conclusions, summary and outlook.

20. One of the strong sides of the text is that it quotes numerous recent references. However, sometimes it is difficult to see the relation between the text and the reference quoted to confirm or illustrate it. Some references contain erroneous data (in ref. 27 “Jayadevimanoranjitham J” should be “Manoranjitham J J”). DOI numbers are missing for all references.

Author Response

Reviewer 3

  1. The novelty of the paper over the state of the art should be better stressed in introduction. This is a very popular and modern topic and a vast number of relevant articles has been published in recent years. This suggestion means that the particular unsolved problem of interest should be fully stated, followed by the proposed approach to overcome them. The paper has its important novelty and the lack of stressing it sufficiently only can work to the authors’ disadvantage.

Answer: Thanks for your kind suggestion. We have added the currently unresolved issues in lines 74-75 as follows:

However, the sensitivity or specificity of current detection methods was insufficient to satisfy food quality evaluation guidelines [33, 34]. In hence, it is necessary to develop novel methods for monitoring food additives with faster, simpler, and highly sensitive properties [35]. In the meanwhile, we also observed that antiseptics always exhibited with strong reduction which could be utilized in reductive donor oxidation-reduction process. And that, it is obvious that ROS possesses high oxidization potential to oxide antiseptics. In addition, this reaction accompanied with a change in the configuration and physical properties of target [36]. In hence, the combined ROS generated properties of shell-isolated colloidal plasmonic nanomaterial-based nanoenzymes for food anti-septics sensing are potential candidate to promote the food antiseptics monitor.

  1. The authors use the term “sensor” to describe their process (nanomaterials and nanoenzyme reactions with the analyte) instead of assigning it to a particular device. Thus “sensing method” would be more appropriate than “sensor”. In a similar vein I also believe that “shell-isolated colloidal plasmonic nanomaterials” would be more up to the point than “shell-isolated colloidal plasmonic cascade reactor”. Please define “sensor” and “reactor”/“nanoreactor” if you use them with specific meanings differing from the generally accepted one. Particularly, I googled “plasmonic cascade reactor” and found zero (!) hits. An identical result has been obtained when searching for “plasmonic cascade nanoreactor”. Usually “sensor” and “reactor” denote devices, not procedures or processes, while “nanoreactor” is a nanostructure serving as a minuscule chemical reactor. It appears that here nanoenzymes and generally core-shell nanoparticles are considered to be sensors and devices on their own.

Answer: Thanks for your kind suggestion. In the revised manuscript, we changed " sensor", "reactor" and "nanoreactor" to "sensing method". Also, " shell-isolated colloidal plasmonic cascade reactor" has been changed to "shell-isolated colloidal plasmonic nanomaterials".

  1. A part of abstract should be rewritten. The sentence “Importantly, with the introduction of DA as a signal molecule in the shell-isolated colloidal plasmonic nanomaterials, which reacted with the oxidation products of the acid preservatives while maintaining the stability of the shell-isolated structure” is unfinished and unclear.

Answer: Thanks for your kind suggestion. In the revised section of the abstract, we have complemented the role of dopamine as follows:

Importantly, with the introduction of signal molecule DA, the cascade reactor exhibited also induced a Raman signal changed by reacted with oxidation product (malondialdehyde) which also improved analysis sensitive in advanced.

  1. An additional figure presenting a nanoreactor and a block-scheme of its general design procedure would be beneficial for the overall text clarity.

Answer: Thanks for your kind suggestion. The schematic of nanoenzyme synthesis is provided in following. Firstly, through optimization (Figure S2), Au NPs (45 nm) with stable LSPR properties were finally selected as cores, subjected to etching reactions and obtained the Au@CN- NPs. The CN- etched Au NPs could provide connection sites for metal and ligand as-sembly around Au@CN- NPs surface. Then, NO-, CN- and Fe precursors (Fe2+ and Fe3+) was connected by metal coordination interaction and formed the metal sodium nitro-prusside (SNP) complex. Finally, the obtained shell-isolated colloidal plasmonic nanomaterials were reacted with dopamine to fabricate the nanoenzyme cascade reactor.

Figure S1. Schematic diagram of the nanoenzyme preparation process

  1. The manuscript is lacking a more explicit description of the physics of localized plasmon influence to the performance of the proposed nanoreactor (primarily extreme light localization, but also hot electron effects and thermoplasmonic effects should be mentioned) within the context of the designed catalytic and SERS activities.

Answer: Thanks for your kind suggestion. We will provide additional insight into the effect of the LSPR effect of the nanoenzyme on catalytic and SERS activity.

“…the nanoenzyme sensor-H2O2-TMB system showed significant color response and UV-visible spectral behavior; its absorbance intensity (at 652 nm) was 12 times higher (Figure S4) than that of the H2O2-TMB reaction system. In addition, no ROS were generated and no obvious color changes was noticed for the other reaction systems. These results were attributed to the ability of the nanoenzymes with a high degree of localization and plasmonic effects to react with H2O2 to produce a high Fenton-like catalyticity, which…”

 “…and the oxidation of SA molecules occurred sequentially. The highly localized, hot-electron and thermal-plasmon effects of the LSPR effect resulted in a dramatic enhancement of the Raman signal while molecule located around nanomaterials. When the signal molecule DA was introduced in the nanoenzyme, the 'fingerprint' SERS signal was enhanced (at 578, 813, 1258, 1313, 1419 and 1463 cm-1), which can use for the validation of the MDA reaction. The generated MDA eventually combined with the DA molecules in Au@SNF NPs and resulted in prominent changes to the SERS signals…”

  1. The authors repeatedly use the expression “double wing with one body” without explaining it. I guess they meant that their structures can serve two purposes at the same time, namely catalytic/nanoenzymatic and localized plasmonic (SERS-enabling). However, a definition should be explicitly written.

Answer: Thanks for your kind suggestion. In the revised manuscript, we have explicitly defined "double wing with one body". Firstly, Au@SNF NPs were developed as a SERS substrate with an internal standard, which could correct for Raman signal fluctuations to improve the stability of quantitative detection. Secondly, Au@SNF NPs bond to DA and participate in Fenton-like reactions, ultimately generating •OH to facilitate the reduction of sorbic acid. Therefore, the sensing method developed based on Au@SNF core-shell nanoparticles constitutes a "double wing with one body" detection strategy.

  1. Almost the whole last paragraph of Introduction actually gives a summary of the paper, which would be much better suited to Conclusion. It is more customary to just announce the novelty of the paper and (shortly) the methodology used, followed by a single-paragraph description of the paper organization and structure.

Answer: Thanks for your kind suggestion. We have revised the final paragraph of the Introduction as follows:

In this study, a “double wing with one body” strategy is developed to fabricate reduc-tive food antiseptic sensors using shell-isolated colloidal plasmonic nanomaterials. Au NPs were used as the core of colloidal plasmonic nanomaterials, and co-assembly with ferrous ions (Fe2+), ferric ions (Fe3+), the nitroso (NO-) group, cyanogen (CN-) group, and dopamine (DA) to form the sensors. In the presence of H2O2, large amounts of •OH were generated by Fenton-like reaction and exhibited with excellent peroxidase-like activity, which also catalyzes sorbic acid oxidation to malondialdehyde (MDA). The formed MDA will reacted with DA molecule and induced a Raman signal changed which has a clear linear relationship with the concentration of SA. The present work demonstrated that the nanoenzyme has significantly enhanced sorbic acid sensing performance and can be used for the examination of sorbic acid concentrations in real food samples.

  1. In line 62, the term “internal standards (ISs)” is used without defining it.

Answer: Thank you for pointing out the problem. The internal standards method (ISs) is effective attempt to improve the reproducibility and quantitative SERS analysis by normalize volatility of SERS signals. We have explained the ISs define in the revised manuscript.

  1. Hydroxyl radicals are written in two different manners, as ·OH and •OH. Is that on purpose? Also, gold nanoparticles are written in two different manners, as “Au NP” and “AuNP”.

Answer: Thank you for pointing out the problem. In the revised manuscript, we have unified the hydroxyl radicals to "•OH" and the gold nanoparticles to "Au NP" respectively.

  1. Some sentences are plain unclear. For instance, what does it mean “generate signals through the self-assembly strategy”? Can “using 3,3',5,5' -tetramethylbenzidine (TMB) mimetic activity of peroxidases classical means” be rewritten in a clearer manner? Other examples include “the SNP complex was assembled around the CN- etched AuNPs and interacted with each other around the monodisperse CN- connected sites”, “spectroscopy behaviors were performed”, “strong Fe atom combined properties”. The clarity and impact of the sentence in lines 209-211 would benefit from either additional explicatory text or by a reference. Please rewrite the unclear sentence “The SERS response of 585 cm-1/2185 cm-1 to 10 µM of glucose with ten times.” in Fig. 5 caption. Please clarify “The method could be used for quantitative analysis and applied for real-time quantitative analysis.” Also clarify “the determination of actual food samples.”

Answer: Thank you for pointing out the problem and we apologize for this writing error. The revised sentence was as follows:

“…In this system, the SERS signal primer DA was covalently bonded around Au@SNF NPs through the self-assembly strategy.”

“…The catalytic efficiency of the Au@SNF NPs was analyzed by using the 3,3',5,5'-tetramethylbenzidine (TMB) colorimetric reaction, which is a classical mimetic peroxidase means. The TMB catalytic…”

“…The CN- etched Au NPs could provide connection sites for metal and ligand assembly around Au@CN- NPs surface.”

“…The UV-Vis spectroscopy and Raman spectra were performed to characterize the prepared nanoenzyme cascade reactor…”

“…The coordination through NO-Fe-NO, CN-Fe-CN, and NO-Fe-CN resulted in the formation of SNF structure around AuNPs, which caused by the excellent complexion interaction of Fe atoms.”

“the two localized surface plasmon resonance (LSPR) bands located at 526 nm and 600-800 nm can be assigned to Au NPs and SNF species, respectively.”

“Figure 5…(D) The SERS intensity ratio of 585 cm-1/2185 cm-1 obtained from 10 sets and measured with three times in parallel with 10 mM glucose solution.”

“…The Au@SNF-based nanoenzyme reactor can solve the problem of low reproducibility of the SERS substrate and meet the requirements of quantitative analysis. The method could be used for real-time quantitative analysis in different fields.”

“…these data indicate the application potential of the method in the determination of complicated food samples. In summary…”

  1. Misleading nomenclature for optical spectral properties is used. “UV-visible absorption” is mentioned at 652 nm (e.g. line 284), “UV-vis spectrum” (line 209), “UV-visible spectral behavior” (lines 287, 296), “UV-visible absorption peak” (lines 298, 307, 324, 326, 372, etc.) although the plasmon resonances appear solely in the visible spectrum. A distinction must be made between the material properties (absorption and generally spectral properties which are in the visible) and the equipment and methods used in characterization procedures (which is always UV-vis spectroscopy/spectrophotometry).

Answer: Thank you for pointing out the problem and we apologize for this writing error. In the revised manuscript, we have corrected the misleading nomenclature that had been used and have unified the equipment as "UV-vis spectroscopy" and the method as "UV-vis spectrophotometry".

  1. Some dispersion spectra are given in dependence of wavelength while others are given in dependence on wave vectors. A unified notation would facilitate intercomparison throughout the text for the readers.

Answer: Thanks for your kind suggestion. In order to make the experiment more intuitive for the reader, the dispersion spectra were unified in terms of wave vectors.

  1. The abbreviation SNF (Sodium nitroprusside framework) is only defined at 4th page, after being previously used several times. I do not see that the term TMBox is defined anywhere in the text. All abbreviations must be defined the first time they are used.

Answer: Thanks for your kind suggestion. In the revised manuscript, we defined all abbreviations when they were first used.

  1. As mentioned in the preamble to these suggestions, the text is mostly well written, meaning that its English is for the most part rather fluent. However, there is also a number of mistakes, erroneous and unclear sentences. Besides the mentioned “sentence” from the Abstract (see suggestion #3) and unclear declarations (suggestion #10), examples of sentences that should be corrected include “sufficient to generated additional ROS” (generate?), “Cheng et al. [9]. fabricated” (point before “fabricated” to be deleted?), “which exhibited high ROS generation capability owing to a metal oxidase-mimicking catalytic process Hexahedral Au@AgPt nanoparticles (NPs) with excellent catalytic properties” (point missing before “Hexahedral”?), “, which also exhibited good catalytic” (“. These NPs also exhibited good catalytic”?), “These above materials” is written at the beginning of a paragraph, “catalysis properties” (catalytic?), “their unregulated use can lead to delayed growth and development” (growth and development of what? fetus?), “novel methods that are faster, simpler, and highly sensitive for monitoring food additives” (is this supposed to mean “novel methods for monitoring food additives that are fast, simple, and highly sensitive”?), “susceptible by the environment” (?), “reductive food antiseptic sensors” (reductive sensing of food antiseptics?), “precursors and organic-ligand was connected” (were?), “Fe is primarily present in the interior of the Au@SNF NPs than on the surface”, “the resulted are depicted in Figure” (results?), “in the absence and presence of” (=with and without?), “In Hence”, “as interfering substances” and “as the interfering substance.” These are not the only language errors and the manuscript would benefit from critical reading by a native speaker or language expert.

Answer: Thank you for pointing out the problem and we apologize for this writing error. We will revise the spelling and grammatical errors and the revised sentence will read as follows:

“…The ROS with high peroxidase or oxidase-mimicking catalytic activities in the shell-isolated structure of the nanomaterials are sufficient to generate additional ROS such as hydroxyl radicals (•OH) and oxygen radicals (•O2-) via hydrogen peroxide dissociation [6-8] …”

“…Cheng [9] fabricated a metal organic skeleton-based shell-isolated nanocatalyst and metal-activated cascade system, which…”

“…which exhibited high ROS generation capability owing to a metal oxidase-mimicking cat-alytic process. Hexahedral Au@AgPt nanoparticles (NPs) with excellent catalytic…”

“…. These NPs also exhibited good catalytic reaction…”

“…These shell-isolated colloidal plasmonic nanomaterials always exhibit better catalysis properties by the quickly trans-forming of enzyme matrix to products and accelerating enzymatic kinetics under physio-logically conditions…”

“…These above materials always exhibit better catalytic properties by the quickly trans-forming of enzyme matrix to products and accelerating enzymatic kinetics under physio-logically conditions…”

“…However, their unregulated use can lead to delayed fetal growth and development [24], gastro-helcosis and issues with the digestive system [25], or fetal malformation…”

“…This has necessitated the development of novel methods for monitoring food additives that are faster, simpler, and highly sensitive…”

“…However, the ROS generation properties of various nanoenzymes are different from each other, and shell-isolated or IS molecules are quite susceptible to the assay microenvironment.”

“…In this study, a “double wing with one body” strategy was developed to fabricate reduced food antiseptic sensors using shell-isolated colloidal plasmonic nanomaterials…”

“…Firstly, NO-, CN- and Fe precursors (Fe2+ and Fe3+) were connected by metal coordination interaction and formed the metal sodium nitroprusside (SNP) complex…”

“…Fe ions were primarily present in the interior of the Au@SNF NPs than on the surface…”

“…Efficiency of the Fenton-like reaction with different parts of the nanoenzyme reactor was investigated, shown in Figure…”

“…We also investigated the UV-visible absorption properties and colorimetric responses of the nanoenzyme reactor in the presence of DA…”

“…SA is an important food antiseptic, which used extensively in food preservation processing. Also, the antiseptics-related reduction behaviors are worthy of attention…”

“…different types of food preservatives, such as sodium dihydrogen phosphate, sodium acetate, benzoic acid, p-hydroxybenzoic acid, and potas-sium metabisulfite, were added as interfering substances”

“…were added as interfering substances. The interfering substances (10-4 M) signal was similar to that of the blank solution…”

  1. The style of the text could be improved by simplifying and rearranging it. A contribution of a language expert would be beneficial here too.

Answer: Thanks for your kind suggestion. We worked on the manuscript for a long time and repeatedly adding and removing sentences and sections apparently. We improved the readability of the document by simplifying and rearranging while we were really hoping for a substantial improvement in the flow and language level.

  1. Fig. 3A, B, C uses sans serif type of font, while D uses serif font. Kindly use sans serif font (Arial? Helvetica?) in 3 D too.

Answer: Thanks for your kind suggestion. In the revised manuscript, we corrected the font in Figure 3D by choosing arial consistently as follows:

Figure 3. (A) the UV-Vis absorption spectral and color changes of the reaction systems: (a) with-out H2O2, (b) without TMB, (c) without nanoenzyme reactors, (d) nanoenzyme reactors-H2O2-TMB. (B) the UV-Vis absorption spectral response and color changes of the Fenton-like reaction: (a) without H2O2, (b) without TMB, (c) Fe3+-H2O2-TMB, (d) Fe3+-DA-H2O2-TMB. (C) The steady-state kinetic assay of Au@SNF NPs-catalyzed oxidation of TMB by H2O2. (D) The mecha-nism of the peroxidase-like catalytic ability with Au@SNF NPs.

  1. Kindly provide reference(s) for the sentence in lines 333-335.

Answer: Thanks for your kind suggestion. In the revised manuscript, we have added the corresponding references to the sentences in lines 333-335 as follows:

[39] Zhang J, Xu Q, Pei W, et al. Self-assembled recombinant camel serum albumin nanoparticles-encapsulated hemin with peroxidase-like activity for colorimetric detection of hydrogen peroxide and glucose. International Journal of Biological Macromolecules, 2021, 193: 2103-2112. 10.1016/j.ijbiomac.2021.11.042

[40] Zhou X, Fan C, Tian Q, et al. Trimetallic AuPtCo nanopolyhedrons with peroxidase-and Catalase-Like catalytic activity for Glow-Type chemiluminescence bioanalysis. Analytical Chemistry, 2021, 94(2): 847-855. 10.1021/acs.analchem.1c03572

  1. Table 1 is unreferenced in the text.

Answer: Thank you for pointing out the problem and we apologize for this error. We have revised the analytical presentation of the actual sample results as follows.

“…As shown in Table 1, the SA content of the actual samples were determined to be less than the maximum usage limit of 1.0 g/kg for confectionery, which was in accordance with the national limit (GB 5009.28-2016 determination of benzoic acid, SA, and sodium saccharin in food)”

  1. Conclusion appears unfinished. It reiterates the text written at the end of Introduction and skips the real conclusions, summary and outlook.

Answer: Thanks for your kind suggestion.

In summary, we proposed an oxidation induced reaction for the detection of food preservatives SA in complex food products with SERS. The shell-isolated colloidal plasmonic nanomaterials Au@SNF NPs-DA system was selected as nanoenzyme to generate •OH and •O2- with Fenton-like reaction. The •OH generated in the cascade reactor had a high oxidative capacity toward acid preservatives, especially for SA. The formed MDA will reacted with DA molecule and induced a Raman signal changed which has a clear linear relationship with the concentration of SA. The as-mentioned method exhibited with highly sensitive and satisfactory recoveries for SA determination in real food sample assays. This novel sensing method has great potential for the quantitative analysis of preservatives in complex food products. In the meanwhile, the use of IS molecules in the shell-isolated structure was effective in realizing a reproducible and quantitative SERS analysis method, which overcame previous limitations and could extend the use of nanoenzymes to various complex sensing applications.

  1. One of the strong sides of the text is that it quotes numerous recent references. However, sometimes it is difficult to see the relation between the text and the reference quoted to confirm or illustrate it. Some references contain erroneous data (in ref. 27 “Jayadevimanoranjitham J” should be “Manoranjitham J J”). DOI numbers are missing for all references.

Answer: Thank you for pointing out the problem and we apologize for this error. In the revised references, we will add DOI numbers and correct erroneous data. The revised references are as follows:

References:

[26] Tungkijanansin N, Alahmad W, Nhujak T, et al. Simultaneous determination of benzoic acid, sorbic acid, and propionic acid in fermented food by headspace solid-phase microextraction followed by GC-FID. Food Chemistry. 2020, 329: 127161. 10.1016/j.foodchem.2020.127161.

[27] Manoranjitham J J, Narayanan S S. Electrochemical sensor for determination of butylated hydroxyanisole (BHA) in food products using poly O-cresolphthalein complexone coated multiwalled carbon nanotubes electrode. Food Chemistry. 2020, 342: 128246. 10.1016/j.foodchem.2020.128246.

[28] Peng L, Yang M, Zhang M, et al. A ratiometric fluorescent sensor based on carbon dots for rapid determination of bisulfite in sugar. Food Chemistry. 2022, 392: 133265. 10.1016/j.foodchem.2022.133265.

Round 2

Reviewer 2 Report

Satisfactory revision.

Reviewer 3 Report

The authors have responded to all of my concerns and improved their manuscript accordingly. I am satisfied with the quality of work they have done. A minor quibble I have with their response to my suggestion 4 is that I believe that Fig. S1 should have been incorporated into the main body of the manuscript instead of being a part of the supplementary files. Albeit that would be clearer and easier for the reader to follow (and I personally would prefer suh a manuscript), their action does represent a valid correction, so I believe that the incorporation of Fig. S1 into the main manuscript should remain optional with the authors. Everything that is left besides that are some small English corrections (e.g. “based strong SERS” instead of “based on strong SERS”, “In hence,” etc.), of the kind that is normally done during the regular editorial production procedure. Therefore I propose the acceptance of the revised paper (the inclusion of Fig. S1 being strongly suggested but not mandatory).